# College students' stress and health in the COVID-19 pandemic: The role of academic workload, separation from school, and fears of contagion

**Chunjiang Yang** *, **Aobo Chen, Yashuo Chen**

School of Economics and Management, Yanshan University, Qinhuangdao, Hebei, China

* ycj@ysu.edu.cn

**Data Availability Statement:** All relevant data are within the manuscript and its Supporting Information files. In addition, all data are available

## Abstract

The COVID-19 pandemic has unhinged the lives of people across the globe. In particular, more than 30 million Chinese college students are home-schooling, yet there is little understanding of how academic workload, separation from school, and fears of contagion lead to a decrease in their health. This study examined the relationships between Chinese college students' three critical stressors and two types of health in the COVID-19 pandemic context. We used a three-wave lagged design with a one-week interval. All the constructs were assessed by self-report in anonymous surveys during the COVID-19 pandemic. College students were asked to report their demographic information, academic workload, separation from school, fears of contagion, perceived stress, and health. The results of this study showed that academic workload, separation from school, and fears of contagion had negative effects on college students' health via perceived stress. In the COVID-19 crisis, multiple prevention and control measures focusing on college students may lead them to have different degrees of stress and health problems. Our results enrich the literature on stress and health and offer novel practical implications for all circles of the society to ensure students' health under the context of the COVID-19 epidemic.

## Introduction

Compared with other student groups, such as primary school students and middle school students, the traditional view is that college students bear more pressure and have more serious physical and mental health problems [1]. In previous research, there is a strongly held consensus that dealing with intimate relationships, financial difficulties, and fulfilling responsibilities and roles are the main sources of stress for college students. Due to the recent social changes in the education domain (e.g., the sharing of educational resources and advances in communication technology), the use of distance education is more and more, which changes the communication patterns between teachers and students, increases the isolation and independence of students, and thus becomes an important source of pressure for students [2]. Today, the spread of coronavirus disease 2019 (COVID-19) is becoming unstoppable, having infected more than

form the OSF database. The link is https://osf.io/ajfhw/ and the DOI is 10.17605/OSF.IO/AJFHW.

**Funding:** National Natural Science Foundation of China (71572170). The funders had no role in study design, data collection and analysis, decision to publish, or preparation of the manuscript.

**Competing interests:** The authors have declared that no competing interests exist.

12 million people [3]. In response to this unprecedented challenge, the Chinese government has ordered a nationwide school closure as an emergency measure to prevent the spreading of the infection among teachers and students. As a consequence, 30.315 million Chinese college students are trapped at home, learning online courses through the internet to complete the required academic tasks. So far, the new virtual semester has been going on for nearly three months, and various courses are offered online in a well-organized manner. Although these decisive actions and efforts are highly commendable and necessary, there are also reasons to worry that drawn-out school suspension, home confinement, and distance learning may have adverse effects on college students' physical and mental health [4]. In addition, a series of issues, such as fear of contagion, frustration and boredom, inadequate information, and lack of private space at home, would continue to emerge and increase during the COVID-19 outbreak [4]. However, previous research failed to give enough attention to college students in the epidemic context. For example, some studies have shown that large-scale isolation measures and loss of income have led to mental health problems among migrant workers during COVID-19 outbreaks [5]. Chen et al. [6] have discussed the problems in psychological intervention services of medical workers. The present study aimed to fill the void by focusing on college students to explore the influence of several important stressors on their health during the COVID-19 outbreak. Specifically, we identified three important stressors among college students—academic workload, separation from school, and fear of contagion and further explored the mechanism behind the relationships between three stressors and mental and physical health.

Academic stressors refer to any academic demands (e.g., environmental, social, or internal demands) that cause a student to adjust his or her behavior. Learning and examination, performance competition, especially mastering much knowledge in a short time, would lead to different degrees of academic pressure [7]. Although all planned courses have been affected by the COVID-19 epidemic, online learning still leaves college students with the same academic burden as usual. In addition, previous evidence shows that separation anxiety disorder of adults is similar to that of children and adolescents in phenomenology. College students who are attached to their classmates may experience separation anxiety after leaving school. Emerging problems during the COVID-19 outbreak, such as conflicting family schedules, changes in eating and sleeping habits, separation from classmates, and loneliness, may have adverse effects on college students [8]. Seligman and Wuyek suggested that college students may experience separation anxiety when they go home for the holidays [9]. Finally, the COVID-19 pandemic has projected humanity into an unprecedented era characterized by feelings of helplessness and loss of control. As Sontag [10] noted, unknown diseases cannot be totally controlled and thus are often considered more threatening than factual evidence. During this period, populations remained almost entirely susceptible to COVID-19, causing the natural spread of infections to exhibit almost perfect exponential growth [11]. Therefore, feelings of fear and apprehension about having or contracting COVID-19 may be a significant stressor for college students. Considering the actual situation of college students during an outbreak of COVID-19, the first aim of the study is to identify academic workload, separation from school, and fears of contagion, which are the three important stressors of college students.

Moreover, previous studies demonstrated that diseases (e.g., SARS) [12], academic (e.g., academic expectations) [13], and attachments (e.g., attachment to parents) [14] are closely associated with students' health. However, it is still not fully understood how these stressors lead to health-related outcomes. In particular, we know little about the mechanism through which these stressors affect physical and mental health in the context of the COVID-19 outbreak. Brewster et al. [15] have suggested that further research is needed to establish the different mechanisms through which stressors impact health in order to have a profound

understanding of the nature of stress. In response to these calls for further research, the second aim of this study thus was to examine the potential mediating roles of perceived stress in the relationships between the three stressors and physical and psychological health.

Overall, the current study focuses on college students and presents an integrated framework to investigate whether and how three types of stressors influence physical and mental health. Our study contributes to the current literature on stress and health in three unique ways. First, to the best of our knowledge, this study is the first to explore home-schooling college students' physical and mental health during the COVID-19 outbreak. It makes up for the lack of understanding of the situations of college students who are ongoing home-schooling during the COVID-19 outbreak. Second, we discovered that academic stressors, interpersonal stressors, and environmental stressors of college students are critical factors that influence their health. This investigation not only overcomes the previous analysis focusing primarily on the effect of a single stressor on college students' health but also enlarges the scope of current research on students' stress and health by shifting the locus of theorizing away from campus domain to family domain. Third, by examining the mediating role of perceived stress, we exposed the "black box" in the relationship between stressors and health.

## Three stressors among college students during the COVID-19 outbreak

College students' stressors have been typically grouped into three major categories: academic pressure [16–19], social and interpersonal pressure [20, 21], and environmental pressure [22, 23]. Specifically, this study focuses on academic workload (representing academic pressure), separation from school (representing social and interpersonal pressure), and fear of contagion (representing environment pressure).

**Academic workload.** Academic problems have been regarded as the most common stressor for college students [24, 25]. For example, in Schafer's [26] investigation, students reported that the most significant daily hassles were academics-related stressors such as constant study, writing papers, preparing for exams, and boring teachers. The academic pressure easily comes from taking and preparing for exams, grade level competition, and acquiring a large amount of knowledge in a short period of time [7]. Perceived stress is a response to stressors, referring to the state of physical or psychological arousal [27]. College students experience adverse physical and psychological outcomes when they perceive excessive or negative stress. Excessive stress may induce physical impairments, including lack of energy, loss of appetite, headaches, or gastrointestinal problems [28]. Numerous studies have evaluated academic stress associated with various adverse outcomes, such as poor health [29, 30], anxiety [31], depression [32], and poor academic performance [33, 34]. In particular, Hystad et al. [35] found significant associations between academic stress and health, both psychological and physical.

**Separation from school.** Previous studies have shown that students view the transition from high school to college as a source of some degree of stress and emotional dissonance [36]. College students' adaptability is closely related to their attachment with different people, such as parents and friends. Indeed, extant research on attachment has four research directions. First, the most systematic studies have focused on adolescents' psychological separation from parents, suggesting that the healthy development of adolescents depends largely on psychological separation from parents [37]. The second field focuses on the attachment relationship between parents and adolescents, assuming that attachment to parents is a necessary prerequisite for adolescents' adaptive function [38, 39]. The third research stream emphasized the importance of psychological separation and attachment. According to this point of view, a balanced parent-child relationship between psychological separation and parental attachment is

the best choice for students' development [37, 40]. Finally, a fourth research stream focuses on attachment to a group or group members in a campus environment and suggests that this attachment may influence students' affective and behavioral outcomes [41]. From the above-mentioned research fields, we can discover that the existing literature mainly focuses on the effects of attachment on college students in their campus life. Of course, it is undeniable that the campus is the primary field for college students, while the family field becomes more important for college students during the weekends, holidays, and internships. Unfortunately, there is a dearth of research on the influence of attachment on college students in their family life. According to the literature on attachment and separation, attachment relationships between college students and their classmates may affect their stress and health when they are at home.

Social cohesion theorists believed group formation was entirely a function of individual relationships among the group members: As individuals were attracted to one another, they were consequently attracted to the group as a whole. Furthermore, group formation can occur independently of interpersonal attraction [42]. Based on the social identity theorists' research, in minimal groups, people formed group attachment without having any contact or even knowing the other members in their group [41]. Thus, whether or not there is a close relationship with group members, one can feel the attachment to the group to which he belongs. However, some scholars reasoned that students with a stronger and healthier sense of themselves as individuals would be better equipped to handle the demands for independent functioning that accompany the college transition. An investigation among medical students shows that students with greater group cohesion reported less stress [43]. Separation anxiety begins with separation from parents, peers, and other significant persons. The national mandate force college students to separate from school in the COVID-19 pandemic, leading to separate from their peers and thus may cause their stress.

**Fears of contagion.** Fears of contagion reflect feelings of apprehension about having or contracting COVID-19. The literature on health anxiety [44] suggests that threatening events —such as a global pandemic—trigger high levels of stress. Although previous studies have described the pressures triggered by large-scale events, such as natural disasters [44] and terrorist attacks [45], the outbreak of infectious diseases (the COVID-19 pandemic) worldwide is different from other large-scale events [46]. Update to *1 May 2020*, the COVID-19 pandemic has resulted in 3175207 confirmed cases and 224172 deaths [3]. To prevent the spread of infectious diseases, Chines schools have been closed nationwide, collective activities have been canceled, public transportation has been suspended, and family imprisonment has been strictly enforced in some epidemic areas. The outbreak of COVID-19 has a seriously destructive impact on people's lives all over the world. The effects of COVID-19 are multifaceted, affecting both physical health (e.g., pneumonia, liver, and renal injury) and psychological well-being (e.g., fear of contracting an infectious disease, avoid exposure to others to reduce the risk of being infected). Although national preventive measures can slow the spread of the epidemic, individuals cannot ensure that they are not infected with the disease. According to Chinese data, a large number of transmissions, both in nosocomial and community settings, occurred through human-to-human contact with individuals showing no or mild symptoms [47]. The COVID-19 pandemic is an immediate threat, it is unclear how long it will persist, and there are a multitude of unanswered questions regarding its impact. Several anecdotal reports by health care professionals note the COVID-19 pandemic triggers individuals' anxiety and stress, particularly surrounding the uncertainties brought by COVID-19 [48]. Therefore, we suggest that fear of having or contracting COVID-19 may lead to college students' stress and health problems.

## Transactional models of stress and coping

Lazarus and Folkman [49] defined stress as a particular relationship between the person and the environment that is appraised by the person as taxing or exceeding his or her resources and endangering his or her well-being. The Transactional Model of Stress provides a basic framework for explaining the processes of individuals coping with stressful events. A core tenet of this model is that the interaction between the person and the environment creates an individual's feeling of stress. When faced with a stressor, two appraisals were triggered, an individual evaluates potential threats or harms (primary appraisal), as well as the ability to change the situation and control negative emotional reactions (secondary appraisal). Primary appraisal is an individual's estimate of the significance of an event as stressful, negative, positive, controllable, challenging, friendly, or irrelevant. The secondary appraisal is a judgment of an individual's coping resources and choices [50]. The response described in the transaction model starts after the primary and secondary appraisals [49]. There are two kinds of coping efforts strongly related to a person's cognitive appraisal. One is referred to as problem-focused coping, and the other is referred to as emotion-focused coping. The former coping strategy has been shown to be more commonly used in a person's causal analysis, suggesting that some measures can be taken to change the negative situation. For example, Folkman and Lazarus [51] show that students will adopt a problem-based coping style before the examination. The latter coping strategy predominates when people assess that they have no options or lack resources to alter the situation, stressors have to be accepted [52]. Students have reported they often use escape/avoidance ways to cope with stressors. The problem-solving efforts try to change the situation actively, while the emotion-centric coping style will only change the individual's interpretation of the stressors. Based on these arguments, we suggest that college students who feel stressed (e.g., regard their stress as an environmental source or lack of ability to alter) would adopt emotion-focused coping as a means of reappraising an uncontrollable situation. Thus, perceived stress plays a central role in the attribution–secondary appraisal coping relationship.

## The present study

Our primary aim in the study was to examine the influence of stressors on college students' stress and health during the COVID-19 outbreak. Specifically, we empirically examine the influence of academic workload, separation from school, and fears of contagion on college students' psychology and physiology health that included perceived stress as a mediator. Based on the literature review, we hypothesized:

H1a: Academic workload is positively correlated with perceived stress.

H1b: Academic workload is negatively correlated with physical and mental health.

H2a: Separation from school is positively correlated with perceived stress.

H2b: Separation from school is negatively correlated with physical and mental health.

H3a: Fears of contagion are positively correlated with perceived stress.

H3b: Fears of contagion are negatively correlated with physical and mental health.

H4: Perceived stress is negatively correlated with physical and mental health.

H5: Perceived stress mediates the relationship between academic workload and physical and mental health.

H6: Perceived stress mediates the relationship between separation from school and physical and mental health.

H7: Perceived stress mediates the relationship between fears of contagion and physical and mental health.

The hypothesized model is presented in Fig 1.

## Methods

### Ethics statement

All participants were treated following the American Psychological Association ethical guidelines, and the study protocol was approved by the Yanshan University Institutional Review Board (Approval no. 20-03-01). Each participant got an introduction and a link to a questionnaire via WeChat, a widely used instant communication tool in China. In the introduction, anonymity, the aim of the survey, and information on whether participants agreed to participate in the survey were all illustrated in detail. We explained that if you want to participate in the survey, please open the questionnaire link and fill in your true feelings. Some college students expressed their intention to participate in written form through WeChat, and other college students directly participated in the survey with tacit approval according to the instructions. Therefore, all participants completed the questionnaire voluntarily. In addition, participants were undergraduate students recruited from four public universities in China. Thus, participants are adults and not minors, and we do not need to obtain consent from parents or guardians.

### Procedure and participants

Our work is situated within an environmental context the COVID-19 pandemic. All variables were assessed with participants' self-reports in three anonymous online surveys. Data were collected in three waves to minimize common method bias [53]. In the first-wave survey (Time 1), participants were asked to report their academic workload, separation from school, fears of contagion, and necessary demographic information. In Time 1, 1072 completed

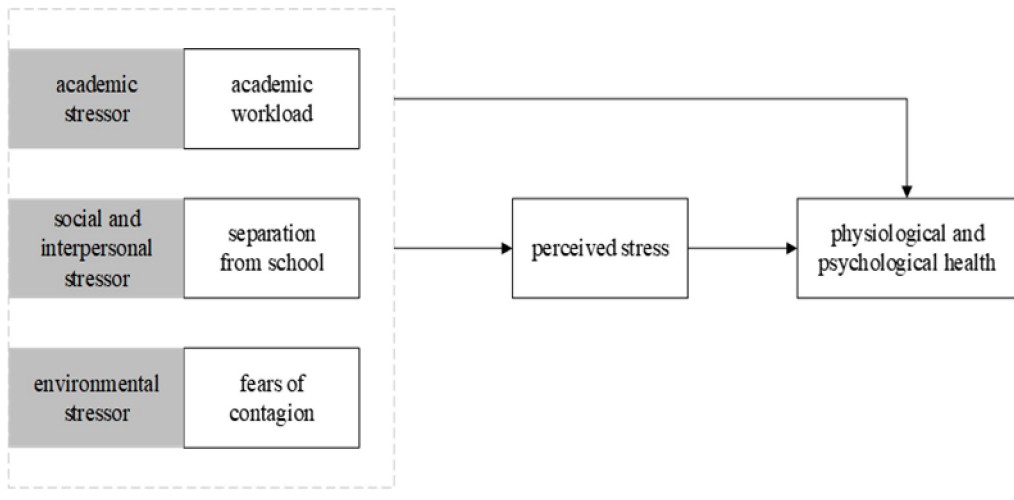

**Fig 1. Hypothesized model.**

questionnaires were returned. One week later (Time 2), students evaluated their level of perceived stress. Time 2 questionnaires were distributed to the 1072 students, with 945 completed questionnaires being returned. Finally, one week after the second-wave survey (Time 3), students evaluated their physical and psychological health. At Time 3, 867 complete questionnaires were returned.

The average age of college undergraduates was 20.17 years, with 69 percent female. There are 348 freshmen, accounting for 40.1%; 200 sophomores, accounting for 23.1%; 305 junior students, accounting for 35.2%; 14 senior students, accounting for 1.6%. The average number of online courses is 8.41. Only 29.2% of the students rated the quality of online courses as good or very good. Nearly half of the participants said they wanted or very much wanted the school to start soon (49.2%). Of the 867 college students who had participated, 56.2% of them were concerned or very concerned about the possibility of contracting COVID-19 after the semester began. The demographic profile of the survey participants is presented in Table 1.

## Measures

We translated the measures from English to Chinese following the commonly used translation/back-translation procedure. All measures are reported in S1 Appendix. Unless otherwise noted, participants responded to all items on a 7-point scale (from 1 = strongly disagree to 7 = strongly agree).

**Academic workload.** It was measured using the three-item scale developed by Hystad et al. [35]. Items included: "I am spending a lot of time thinking about how this semester's grades could negatively affect my educational and career goals," "I am worrying a great deal about the effect this semester's grades will have on my future," and "I find myself very concerned about the grades I am likely to receive this semester." Cronbach's alpha in their study was .85, which suggests that this scale has good reliability. In the current study, Cronbach's α of this variable is .883.

**Separation from school.** Separation from school was measured using the attachment avoidance scale developed by Smith et al. [54]. The scale has fifteen-item. An example of a

**Table 1. The demographic profile of the survey participants.**

| Variables | Frequency | Percentage |
|---|---|---|
| **Age** | | |
| 17–20 | 534 | 61.6% |
| 21–24 | 328 | 37.8% |
| 24–30 | 5 | 0.6% |
| **Gender** | | |
| Female | 598 | 69.0% |
| Male | 269 | 31.0% |
| **Grade** | | |
| Freshman year | 348 | 40.1% |
| Sophomore year | 200 | 23.1% |
| Junior year | 305 | 35.2% |
| Senior year | 14 | 1.6% |
| **Number of online courses** | | |
| 1–5 | 89 | 10.3% |
| 6–10 | 599 | 69.1% |
| 11–15 | 171 | 19.8% |
| 16–20 | 7 | 0.8% |

reworded scale item is "I find it difficult to allow myself to depend on my group." Factor analytic results in their research suggested that the scale has good reliability. In our current study, we reworded to refer to participants' classmates rather than their social group. An example of a reworded scale item is "I find it difficult to allow myself to depend on my classmates." Cronbach's α of this variable is .929.

**Fears of contagion.**   We developed a six-item scale to assess participants' fears of COVID-19 infection. Six items included "In public, I don't care about touching the door handle without protection," "In public, I don't mind sitting in a chair that has just been sat on," "In an elevator, I don't mind pushing a button without protection," "When I'm in a crowded place, I don't worry about coronavirus from other people," "I don't worry about infection if other people don't wear masks," "Wearing a mask would make me feel safe." In the current study, Cronbach's α of this variable is .842. The results of confirmatory factor analysis are shown that $\chi 2 =$ 74.424; $df$ = 8; $RMSEA$ = .098; $CFI$ = .982; $TLI$ = .966; $SRMR$ = .043. Thus, this scale has acceptable reliability and validity.

**Perceived stress.**   The 10-item Perceived Stress Scale [55] was used to measure the student's stress level in the past month. Participants responded to the items on a 5-point scale (from1 = never to 5 = very often). A sample item is "In the last month, how often have you felt nervous and "stressed"?" In our research, Cronbach's α of this variable is .792.

**Physical and psychological health.**   The CHQ-12 of the Chinese version was used to measure physical and psychological health. The CHQ-12 has been widely used in Chinese populations. The 12 items included headaches, heart palpitations, chest pain or tightening, trembling or pins and needles, sleeplessness, nervousness, and hopelessness. Participants responded to the items on a 4-point scale (1 = not at all, 4 = more than usual). A higher score represented a more severe psychosocial impairment. In our research, Cronbach's α of this variable is .895.

**Control variables.**   We controlled students' age, gender, grade, and the number of online courses in the study.

**Analysis strategy.**   First, we conducted a series of confirmatory factor analyses (*CFA*) in order to test the discriminant validity of the five prime constructs (academic workload, separation from school, fears of contagion, perceived stress, and health). CFA, as an empirical research technology, is affiliated with structural equation modeling. Therefore, it is necessary to judge the fitting situation according to the fitting indexes from the structural equation model. Common fitting indexes include chi-square($\chi 2$), degree of freedom (*df*), *CFI*, *TLI*, *RMSEA*, and S*RMR*. Specifically, if the ratio of chi square to degree of freedom is less than 5, the model is generally acceptable. When *CFI* and *TLI* are higher than 0.9, the model fits well. The smaller the *RMSEA* and *SRMR*, the better the result, and in particular, when it is below 0.08, the model is acceptable. Second, we performed Harman's single-factor test to explore the potential influence of common method variance. No single factor accounting for more than 50% of the variance of all the relevant items indicates that the results are acceptable. Third, we calculated Pearson's correlation coefficient using SPSS Version 21, which reflects the effect of change in one variable when the other variable changes. Fourth, we tested our hypotheses using a path analysis in Mplus Version 8.3. To examine mediation (Hypothesis 5,6, and 7), we used a bootstrap simulation with 1,000 replications to create our bias-corrected 95% confidence intervals (CIs) around our indirect effects. The bootstrap approach is a more robust strategy than the causal step procedure for small samples for assessing indirect effects and a useful method for avoiding power problems relating to a non-normal sampling of the indirect effect. When the 95% confidence interval of the path coefficient does not contain zero, the mediating effect is significant.

# Results

## Confirmatory factor analyses

Table 2 presents the CFA results. As shown, the baseline five-factor model fitted the data well ($\chi2$ = 3020.006; $df$ = 965; *RMSEA* = .050; *CFI* = .920; *TLI* = .914; *SRMR* = .064). Against this baseline five-factor model, we tested four alternative models: model 1 was a four-factor model with academic workload merged with perceived stress to form a single factor ($\chi2$ = 5405.203; $df$ = 969; *RMSEA* = .073; *CFI* = .827; *TLI* = .815; *SRMR* = .112); model 2 was three-factor model with separation from school merged with academic workload and perceived stress to form a single factor ($\chi2$ = 6756.895; $df$ = 972; *RMSEA* = .083; *CFI* = .774; *TLI* = .760; *SRMR* = .096); model 3 was a two-factor model, with fears of contagion merged with separation from school, academic workload, and perceived stress to form a single factor ($\chi2$ = 9752.729; $df$ = 974; *RMSEA* = .102; *CFI* = .658; *TLI* = .636; *SRMR* = .110) and model 4 was a one-factor model with five constructs merged with one factor ($\chi2$ = 13665.328; $df$ = 975; *RMSEA* = .123; *CFI* = .505; *TLI* = .475; *SRMR* = .143). As Table 2 shows, the fit indexes supported the hypothesized five-factor model, providing evidence of the construct distinctiveness of fears of contagion, separation from school, academic workload, and perceived stress and health.

## Tests for common method variance

Because we collected student's self-report of fears of contagion, separation from school, academic workload, and perceived stress and health in the study, common method variance (CMV) may present as a problem. Therefore, we measured different constructs at different time points to decrease common method variance as much as possible [56] and showed high respect for the security, anonymity, and privacy of research objects and informants. In addition, the results of Harman's single-factor test suggest that an exploratory factor analysis of all items explained 68.86% of the total variance, and the largest factor accounted for only 24.73% of the variance.

## Descriptive statistics

The bivariate Pearson Correlation produces a sample correlation coefficient, which measures the strength and direction of linear relationships between pairs of continuous variables. Reported in Table 3 are the means, standard deviations and bivariate correlations of variables. Academic workload, separation from school, fears of contagion, and perceived stress were all negatively correlated with health ($r$ = -.121, $p$ < .01; $r$ = -.289, $p$ < .01; $r$ = -.242, $p$ < 0.01; $r$ = -.225, $p$ < 0.01, respectively). Academic workload, separation from school, and fear of contagion, were all positively correlated with perceived stress ($r$ = .152, $p$ < .01; $r$ = .207, $p$ < .01; $r$ =

**Table 2. Comparison of measurement models.**

| Model | Factors | $\chi^2$ | df | $\Delta\chi^2$ | RMSEA | CFI | TLI | SRMR |
|---|---|---|---|---|---|---|---|---|
| Baseline model | Five factors | 3020.006 | 965 | | 0.050 | 0.920 | 0.914 | 0.064 |
| Model 1 | Four factors | 5405.203 | 969 | 2385.197** | 0.073 | 0.827 | 0.815 | 0.112 |
| Model 2 | Three factors | 6756.895 | 972 | 3736.889** | 0.083 | 0.774 | 0.760 | 0.096 |
| Model 3 | Two factors | 9752.729 | 974 | 6732.723** | 0.102 | 0.658 | 0.636 | 0.110 |
| Model 4 | One factor | 13665.328 | 975 | 10645.322** | 0.123 | 0.505 | 0.475 | 0.143 |

Note: AW = academic workload; SFS = separation from school; FOC = fears of contagion; PS = perceived stress; Two factors = AW+SFS+FOC+PS, Health; Three factors = AW+SFS+PS, FOC, Health; Four factors = AW+PS, SFS, FOC, Health.

.133, $p < 0.01$, respectively). It appears that these findings preliminarily provided support for our hypotheses.

## Hypothesis testing

We performed structural equation modeling (SEM) to test the proposed hypotheses and summarized the standardized values of the path coefficients and their significance levels in Fig 2. The results showed significant positive path coefficients from academic workload to perceived stress ($\beta$ = .211, $SE$ = .040, $p < .01$), providing empirical evidence in support of H1a. Furthermore, academic workload was shown to have a negative effect on health ($\beta$ = -.053, $SE$ = .037, $n.s.$), not providing empirical support for H1b. We also demonstrated that separation from school had a positive association with perceived stress ($\beta$ = .324, $SE$ = .043, $p < .01$) and had a negative association with health ($\beta$ = -.252, $SE$ = .039, $p < .01$), providing empirical evidence in support of H2a and H2b. Moreover, our results showed that fears of contagion were significantly related to perceived stress ($\beta$ = .121, $SE$ = .039, $p < 0.01$) and were significantly related to health ($\beta$ = -.088, $SE$ = .035, $p < .05$), providing empirical evidence in support of H3a and H3b. Perceived stress had a negative effect on health ($\beta$ = -.240, $SE$ = .050, $p < .01$), providing empirical evidence in support of H4.

We further examined the mediating role of perceived stress with nonparametric bootstrapping procedures. As shown in Table 4, the influence of academic workload on health was mediated by perceived stress because the indirect influence of academic workload on health registered the value of b at -.051 ($SE$ = .016, $p$ = .001), with the 95% $CI$ [-.081, -.028]. Therefore, H5 was supported. Similarly, the influence of separation from school on health was mediated by perceived stress because the indirect influence of separation from school on health registered the value of b at -.078 ($SE$ = .020, $p$ = .000), with the 95% $CI$ [-.113, -.049]. Therefore, H6 was supported. The influence of fear of contagion on health was mediated by perceived stress because the indirect influence of fear of contagion on health registered the value of b at -.029 ($SE$ = .012, $p$ = .014), with the 95% $CI$ [-.053, -.013]. Therefore, H7 was supported.

## Discussion

The spread of COVID-19 is becoming unstoppable and has already influenced people and countries all over the world. Holmes et al. [57] called for that multidisciplinary science research must be central to the international response to the COVID-19 pandemic and provide evidence-based guidance on responding to promoting people's health and wellbeing during the COVID-19 pandemic. To answer this call, we focus on college students getting home-

**Table 3. Means, standard deviations, and bivariate correlations among studied variables.**

| Variables | M | SD | 1 | 2 | 3 | 4 | 5 | 6 |
|---|---|---|---|---|---|---|---|---|
| 1. Age | 20.170 | 1.393 | | | | | | |
| 2. Number of online courses | 8.410 | 2.804 | .368** | | | | | |
| 3. Academic workload | 4.900 | 1.363 | .013 | .047 | | | | |
| 4. Fears of contagion | 3.093 | 1.237 | -.040 | -.064 | .009 | | | |
| 5. Separation from school | 3.358 | 1.059 | -.040 | .053 | .014 | .171** | | |
| 6. Perceived stress | 3.064 | .455 | -.013 | .014 | .152** | .133** | .207** | |
| 7. Health | 3.075 | .547 | -.011 | .015 | -.121** | -.242** | -.289** | -.225** |

Note.

** = p < 0.01

* = p < 0.05.

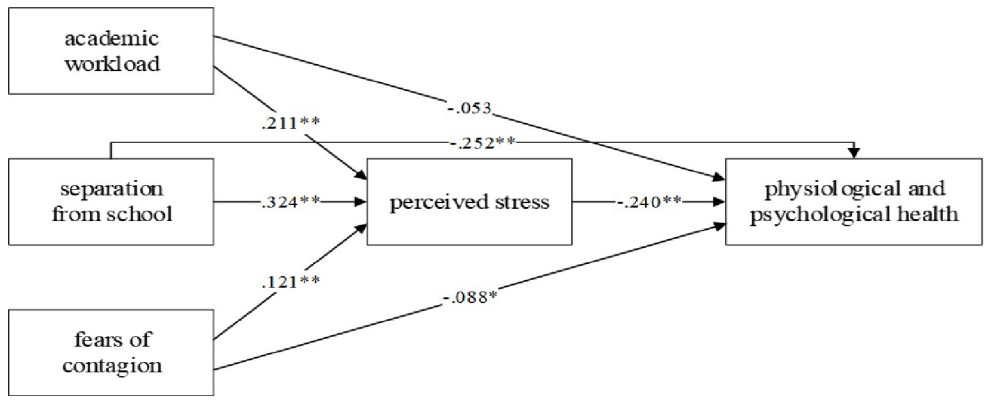

**Fig 2. The result of structural equation modeling.**

schooling to explore their stress and health problems. Although the COVID-19 is still spreading rapidly and widely worldwide, it has been effectively controlled in China. What has happened in China shows that quarantine, social distancing, and isolation of infected populations can contain the epidemic. Whereas individual coping strategies are possible (e.g., social distancing), the spread of the virus at a state level is still beyond any given individual's control. The continuous spread of the epidemic, strict isolation measures, and delays in starting schools, colleges, and universities across the country are expected to influence college students. Considering stress and anxiety associated with the current COVID-19 pandemic for college students [49], it is urgent for the society and management departments to understand the actual situation of college students timely and accurately. Based on the Transactional Model of Stress and coping [49], this study explored the influence of academic workload, separation from school, and fears of contagion on college students' physical and physiological health, as well as the mediating effect of perceived stress in those relationships.

The current study contributes to the existing literature. First, the present study goes beyond previous literature on college students' health during the epidemic by integrating three types of stressors from different fields in the proposed model. As highlighted in previous research on college students' academic stress, preparing exams, courses, and papers should exhibit a negative effect on individual health. During the COVID-19 outbreak, Chinese college students' learning was not suspended, but they attend the various courses offered online follow the regular schedule. While those measures of the virtual semester ensure regular study, they also cause stress on students. In addition, given the importance of social groups to an individual's identity and self-worth, we found that college students were separated from their classmates during the COVID-19 epidemic, which brought them stress and anxiety. Previous evidence suggests that

**Table 4. Results of structural equation modeling on the mediating effect of positive emotions.**

| Indirect paths | Effects | SE | p | Lower 5% | Upper 5% | Results |
|---|---|---|---|---|---|---|
| AW→Perceived stress→ Health | -0.051 | 0.016 | 0.001 | -0.081 | -0.028 | Supported |
| SFS→Perceived stress→ Health | -0.078 | 0.020 | 0.000 | -0.113 | -0.049 | Supported |
| FOC→Perceived stress→Health | -0.029 | 0.012 | 0.014 | -0.053 | -0.013 | Supported |

Note: AW = academic workload; SFS = separation from school; FOC = fears of contagion; We report the 95% confidence intervals (CIs) calculated using 1,000 bootstrap samples, with lower and upper limits in brackets.

college students usually keep attachment relationships with their social group [12, 58]. Attachment figures are usually parents, but may also be siblings, grandparents, or group. Unlike most previous studies that focused on separation from parents [9], this study focused on the influence of departure from school and schoolmates, which is particularly relevant to the epidemic situation. For college students attached to their school or classmates, school-closure is a kind of separation experience, which may be different from their experience when they leave home. Considering the relatively new separation (separation from school) caused by the outbreak of COVID-19, our findings suggest that separation from school was positively related to college students' perceived stress during home-schooling. Finally, we found that exposure to a potentially infectious environment would lead to people's stress, which is in line with previous research that pointed out the negative correlation between the risk of infection and life satisfaction [58]. Similar results have been found in the study of College Students' psychological adjustment during SARS. For example, Main et al. [12] found that the experience of SARS-related stressors was positively associated with psychological symptoms for Chinese college students during the outbreak. Thus, we supposed that during an acute large-scale epidemic such as the SARS and COVID-19 epidemic, even among persons who were not directly contaminated with the disease, the psychological influence of the outbreak on them was significant. In doing so, we identified three important stressors for college students in the COVID-19 pandemic, providing essential inspiration for college students to maintain their physical and mental health during the current epidemic.

Second, based on a transactional model view, we provide a plausible mechanism for explaining the association between academic workload, separation from school, and fears of contagion and health. The transactional model posits that stress responses emerge from appraisal processes that begin when individuals experience a stressor. During primary appraisal, perceptions of elements of the focal stressor are used to determine the degree of threat or harm that this stressor represents; during secondary appraisal, individuals consider if and how they can resolve the underlying stressor. COVID-19, a contagious respiratory illness, is an ongoing, global health crisis, and the greatest challenge we have faced since World War II [59]. The COVID-19 pandemic is a grim but illustrative anxiety-inducing stressor; an uncertain and ongoing threat that cannot be resolved via individual efforts. When individuals have few resources, ways, or abilities at their disposal to deal with the stressors, they generate stress and anxiety and ultimately lead to negative consequences. Thus, perceived stress may be a mediator, transmitting the effects of academic workload, separation from school, and fears of contagion on health-related outcomes. These findings suggest that academic workload, separation from school, and fears of contagion may contribute to youth's general perceived stress, which in turn, may negatively influence their physical and psychological health. Our findings supported Lorenzo-Blanco and Unger's [60] and Sariçam's [61] proposition that perceived stress plays an important role in influencing psychological well-being.

The results of the current study are of significance to practice and policy. First, the administrative department needs to raise the awareness of potential physical and mental health influences of home-schooling during the outbreak. While taking effective prevention measures, the government should guide the mass media to spread positive information and control rumors. Communication can serve as an important resource in dealing with the difficulties of family matters. For instance, parents can share their life experiences with their children and advocate for them to develop good living habits and enjoy a healthy lifestyle. Psychologists can communicate with students by social media platforms to help them cope with mental health issues caused by domestic conflicts, tension with parents, and anxiety from becoming infected. In addition, schools play an important role not only in providing students with educational materials but also in providing students with opportunities to interact with teachers and students.

Schools should also provide guidelines and principles for effective online learning and ensure that content meets educational requirements. Nevertheless, it is also important not to overburden students.

## Limitations

Despite the potential contribution that the present study makes to the mental health field, limitations of the study should be noted. First, a potential limitation is that all measures came from the same source, raising the potential for same-source measurement biases. However, we used a variety of means to reduce this issue, including varying our response scales and separating our measures in time. Further, as we were interested in how college students have dealt with the pandemic over time, focusing on self-reported experiences was appropriate. Another potential limitation of our study is concerned with causality. Hence, future research should conduct the experimental design or utilize the longitudinal data to ensure the conclusion reflects causation. Finally, our research reflects only the impact of the COVID-19 pandemic for college students, and much work is needed to gain a complete understanding of the implications of this crisis for students. Future research could consider how individual factors, such as self-concept, and contextual factors, such as social support, may influence college students' response to perceived stress. Moreover, research focused on within-person fluctuations of perceived stress during this time would also be instructive, as there is no doubt that individuals have experienced considerable variability on a daily basis during the pandemic.

## Conclusions

Confronting the COVID-19 outbreak and variously rigorous measures to prevent the spreading of the infection, college students may feel stress and have more or fewer health problems. Academic workload, psychological separation from school, and fear of contagion were positively associated with the perceived stress and negatively associated with physical and psychological health. In addition, perceived stress is a key mechanism in the relationships between three stressors and two forms of health. This study makes not only unique theoretical contributions to the stress and health literature during the COVID-19 outbreak but also offers novel practical implications for joint efforts from all circles of society to ensure students' health.

## Supporting information

**S1 Appendix.**
(DOCX)

**S1 Data. Anonymized data.**
(XLSX)

**S1 File. Ethical approval.**
(PDF)

## Author Contributions

**Conceptualization:** Chunjiang Yang, Aobo Chen, Yashuo Chen.

**Data curation:** Aobo Chen.

**Formal analysis:** Aobo Chen.

**Funding acquisition:** Chunjiang Yang.

**Investigation:** Chunjiang Yang, Aobo Chen.

**Methodology:** Yashuo Chen.

**Supervision:** Chunjiang Yang.

**Writing – original draft:** Chunjiang Yang, Aobo Chen, Yashuo Chen.

**Writing – review & editing:** Chunjiang Yang, Aobo Chen, Yashuo Chen.

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
