## [Decision Letter · Decision Letter 0]

30 Dec 2020

PONE-D-20-32875

College students’ stress and health during the COVID-19 outbreak: the effects of academic workload, separation from school, and fears of contagion

PLOS ONE

Dear Dr. Yang,

Thank you for submitting your manuscript to PLOS ONE. After careful consideration, we feel that it has merit but does not fully meet PLOS ONE’s publication criteria as it currently stands. Therefore, we invite you to submit a revised version of the manuscript that addresses the points raised during the review process.

One expert in this field has carefully reviewed your submission and he pointed out some merits in your work. However, a major concern regarding the sampling should be tackled in the revision. That says, the samples seemed to be different across time points and the authors should provide clear rationale and how they take care of the methodology issues if they really used different samples. This is a fatal point for me to judge whether the revision can be accepted. 

We look forward to receiving your revised manuscript.

Kind regards,

Chung-Ying Lin

Academic Editor

PLOS ONE

Journal Requirements:

2. Please ensure that you include a title page within your main document.

We do appreciate that you have a title page document uploaded as a separate file, however, as per our author guidelines (http://journals.plos.org/plosone/s/submission-guidelines#loc-title-page) we do require this to be part of the manuscript file itself and not uploaded separately.

3. Please provide additional details regarding participant consent.

In the ethics statement in the Methods and online submission information, please ensure that you have specified (i) whether consent was informed and (ii) what type you obtained (for instance, written or verbal, and if verbal, how it was documented and witnessed). If your study included minors, state whether you obtained consent from parents or guardians. If the need for consent was waived by the ethics committee, please include this information.

4. Please include additional information regarding the survey or questionnaire used in the study and ensure that you have provided sufficient details that others could replicate the analyses.

For instance, if you developed a questionnaire as part of this study and it is not under a copyright more restrictive than CC-BY, please include a copy, in both the original language and English, as Supporting Information, or include a citation if it has been published previously.

5. In the Methods, please discuss whether and how the questionnaire was validated and/or pre-tested. If these did not occur, please provide the rationale for not doing so.

6. We note that you have indicated that data from this study are available upon request. PLOS only allows data to be available upon request if there are legal or ethical restrictions on sharing data publicly. For more information on unacceptable data access restrictions, please see http://journals.plos.org/plosone/s/data-availability#loc-unacceptable-data-access-restrictions.

7. Your ethics statement should only appear in the Methods section of your manuscript. If your ethics statement is written in any section besides the Methods, please delete it from any other section.

Reviewers' comments:

Reviewer's Responses to Questions

**Comments to the Author**

1. Is the manuscript technically sound, and do the data support the conclusions?

Reviewer #1: Partly

2. Has the statistical analysis been performed appropriately and rigorously? 

Reviewer #1: Yes

3. Have the authors made all data underlying the findings in their manuscript fully available?

Reviewer #1: No

4. Is the manuscript presented in an intelligible fashion and written in standard English?

Reviewer #1: Yes

5. Review Comments to the Author

Reviewer #1: In general, this is a good study considering the concept backed by a sound theoretical framework. However, authors need to improve the study’s write-up in several sections of the manuscript especially the English language. I will strongly recommend a native English Language speaker to edit it.

Introduction

Hypotheses

Knowing that we cannot make a causal inference from this study, it will be more prudent to use associate/relate rather than affect.

Methods:

Page 8: Procedure and Participants

Can authors justify why they used different samples for each Time wave and how they resolve the confounding issues arising from this method?

Also, there seems to be a typo in the number of data collected for wave one.

Authors should start a sentence with words and not figures as did with 49.2% and 56.2%.

Measures

Authors may want to revise this section incorporating the number of items per scale, the original developer (author) of each scale, original psychometric property, and how it was scored in this study. Much more importantly, the psychometric properties of these measures for this study.

Data analysis

Authors may end the method section with the above sub-title. This section may detail the types of analytic tools used and the purposes for using them. Cut-off or other criteria related to these analytic tools may be written here.

Results

Tests for common method variance: The first sentence under this sub-section seems incomplete.

Table 3: for bivariate relationships, authors may take out categorical variables from the results as it does not make meaning.

Table 4: Authors may add further details to the table as effect and CI are not enough to get a better picture of the analysis/data.

Discussion

Page 14: “Given the ongoing COVID -19 epidemic to inducing panic, and anxiety [49], therefore, timely and…” this sentence is difficult to understand, please revise.

Authors may try to relate the theory to the findings thoroughly.

Explaining the “SFS” with separation anxiety disorder is too far-fetched (exaggerated and unconvincing). Besides, these are students above 17 year old. SFS have a role to play in the lives of students but not from the angle of separation anxiety but probably as a supporting/coping strategy.

Limitation

The way the authors reported the procedure of the data collection did not reflect a cross-sectional design. Hence, it will make readers doubt the authenticity/validity of this study.

6. PLOS authors have the option to publish the peer review history of their article (what does this mean?). If published, this will include your full peer review and any attached files.

Reviewer #1: No

---

## [Author Response · Author response to Decision Letter 0]

13 Jan 2021

A Point-by-Point Response to Editor Comments

Your comment:

One expert in this field has carefully reviewed your submission and he pointed out some merits in your work. However, a major concern regarding the sampling should be tackled in the revision. That says, the samples seemed to be different across time points and the authors should provide clear rationale and how they take care of the methodology issues if they really used different samples. This is a fatal point for me to judge whether the revision can be accepted.

Our response:

Thank you for giving us the opportunity to revise our paper. We hope that you find this revised manuscript significantly improved and up to your expectation. 

Thank you for your question. Indeed, although the numbers of participants are different across time points, we didn’t use different samples in our research. 

First, we explain what the time-lagged design is and why it is used in our research.

The time-lagged design is a type of research design in which the same person reports information or data of different variables at different time points, which has been widely used in science research (e.g., Laschinger & Fida, 2014; SPENCE LASCHINGER & Finegan, 2008; Kilroy et al., 2017). The time-lagged design can ensure the constructs we concerned would be less influenced by common method bias (e.g., Podsakoff et al. 2003). Generally, the independent variables (IV) are first collected at Time-1, and then the mediation variables (mediators) at Time-2, and finally the dependent variables (DV) at Time-3 from the same person. In our research, academic workload, separation from school, and fears of contagion are independent variables, perceived stress is a mediator, and health is a dependent variable. Thus, we used the time-lagged design and these data of different variables were collected at three time points with one-week interval. 

Second, we describe how to conduct the collection of data and explain that we didn’t use different samples in our study. 

In the first-wave survey (Time 1), participants were asked to report their academic workload, separation from school, fears of contagion, and necessary demographic information, and we got 1072 questionnaires. One week later (Time 2), perceived stress needs to be measured, and we distributed questionnaires to the 1,072 people who had participated in the first-wave survey. However, 127 participants quit the survey and we got 945 questionnaires at Time 2 (valid response rate: 945/1072=88.15%). One week after the second-wave survey (Time 3), physical and psychological health needs to be measured, and we distributed questionnaires to the 945 people who had participated in both the first-wave and second-wave survey. However, 78 participants quit the survey, and we got 867 questionnaires at Time 3(valid response rate: 867/945=91.95%). That is, we got completed data, including the data of all variables used in our research from 867 participants who participated in three surveys. In the data analysis, we used data from only 867 participants. Because some participants did not participate fully in the three surveys, their data was incomplete and thus could not be included in the study. That is, because some participants dropped out of the survey during the data collection process, the number of participants varied at different time points, as shown in Figure 1. 

Reference:

Podsakoff, P. M., MacKenzie, S. B., Lee, J. Y., & Podsakoff, N. P. (2003). Common method biases in behavioral research: a critical review of the literature and recommended remedies. Journal of applied psychology, 88(5), 879.

Laschinger, H. K. S., & Fida, R. (2014). A time-lagged analysis of the effect of authentic leadership on workplace bullying, burnout, and occupational turnover intentions. European Journal of Work and Organizational Psychology, 23(5), 739-753.

SPENCE LASCHINGER, H. K., & Finegan, J. (2008). Situational and dispositional predictors of nurse manager burnout: a time‐lagged analysis. Journal of Nursing Management, 16(5), 601-607.

Kilroy, S., Flood, P. C., Bosak, J., & Chênevert, D. (2017). Perceptions of high‐involvement work practices, person‐organization fit, and burnout: A time‐lagged study of health care employees. Human Resource Management, 56(5), 821-835.

A Point-by-Point Response to Reviewer Comments

Your comment:

1. Is the manuscript technically sound, and do the data support the conclusions?

Reviewer #1: Partly

Our response: 

Thank you very much. Following your suggestions, we carefully revised the manuscript. All changes were marked in blue text. Below please see our point-to-point response to your comments. We hope that you find this revised manuscript significantly improved and up to your expectation. 

Your comment:

2. Has the statistical analysis been performed appropriately and rigorously?

Reviewer #1: Yes

Our response: 

Thank you very much for your recognition and support!

Your comment:

3. Have the authors made all data underlying the findings in their manuscript fully available?

Reviewer #1: No

Our response: 

Thank you very much. We would like to provide and share the data we used in our research. 

Your comment:

4. Is the manuscript presented in an intelligible fashion and written in standard English?

Reviewer #1: Yes

Our response: 

Thank you very much for your recognition and support!

Your comment:

5. Review Comments to the Author

Reviewer #1: In general, this is a good study considering the concept backed by a sound theoretical framework. However, authors need to improve the study’s write-up in several sections of the manuscript especially the English language. I will strongly recommend a native English Language speaker to edit it.

Our response: 

Thank you very much for your recognition and support! We have had a professional copy editor with English as her first language to thoroughly go through the draft. We are confident that the writing has been improved and hope that the revised manuscript will meet your requirements. All changes were marked in blue in the revised manuscript. 

Your comment:

Introduction

Hypotheses

Knowing that we cannot make a causal inference from this study, it will be more prudent to use associate/relate rather than affect.

Our response: 

Thank you very much. Following your suggestion, we use associate/relate rather than affect in the revised manuscript. All changes were marked in blue text. 

Please see 

Page 2 Line 44-Page 8 Line 208:

Page 8 Line 210-Page 9 Line 220:

Our primary aim in the study was to examine the influence of stressors on college students’ stress and health during the COVID-19 outbreak. Specifically, we empirically examine the influence of academic workload, separation from school, and fears of contagion on college students’ psychology and physiology health that included perceived stress as a mediator. Based on the literature review, we hypothesized: 

H1a: Academic workload is positively correlated with perceived stress.

H1b: Academic workload is negatively correlated with physical and mental health.

H2a: Separation from school is positively correlated with perceived stress. 

H2b: Separation from school is negatively correlated with physical and mental health.

H3a: Fears of contagion are positively correlated with perceived stress. 

H3b: Fears of contagion are negatively correlated with physical and mental health.

H4: Perceived stress is negatively correlated with physical and mental health.

Your comment:

Methods:

Page 8: Procedure and Participants

Can authors justify why they used different samples for each Time wave and how they resolve the confounding issues arising from this method?

Our response:

Thank you for giving us the opportunity to revise our paper. We hope that you find this revised manuscript significantly improved and up to your expectation. 

Thank you for your question. Indeed, although the numbers of participants are different across time points, we didn’t use different samples in our research. 

First, we explain what the time-lagged design is and why it is used in our research.

The time-lagged design is a type of research design in which the same person reports information or data of different variables at different time points, which has been widely used in science research (e.g., Laschinger & Fida, 2014; SPENCE LASCHINGER & Finegan, 2008; Kilroy et al., 2017). The time-lagged design can ensure the constructs we concerned would be less influenced by common method bias (e.g., Podsakoff et al. 2003). Generally, the independent variables (IV) are first collected at Time-1, and then the mediation variables (mediators) at Time-2, and finally the dependent variables (DV) at Time-3 from the same person. In our research, academic workload, separation from school, and fears of contagion are independent variables, perceived stress is a mediator, and health is a dependent variable. Thus, we used the time-lagged design and these data of different variables were collected at three time points with one-week interval. 

Second, we describe how to conduct the collection of data and explain that we didn’t use different samples in our study. 

In the first-wave survey (Time 1), participants were asked to report their academic workload, separation from school, fears of contagion, and necessary demographic information, and we got 1072 questionnaires. One week later (Time 2), perceived stress needs to be measured, and we distributed questionnaires to the 1,072 people who had participated in the first-wave survey. However, 127 participants quit the survey and we got 945 questionnaires at Time 2 (valid response rate: 945/1072=88.15%). One week after the second-wave survey (Time 3), physical and psychological health needs to be measured, and we distributed questionnaires to the 945 people who had participated in both the first-wave and second-wave survey. However, 78 participants quit the survey, and we got 867 questionnaires at Time 3(valid response rate: 867/945=91.95%). That is, we got completed data, including the data of all variables used in our research from 867 participants who participated in three surveys. In the data analysis, we used data from only 867 participants. Because some participants did not participate fully in the three surveys, their data was incomplete and thus could not be included in the study. That is, because some participants dropped out of the survey during the data collection process, the number of participants varied at different time points, as shown in Figure 1. 

Reference:

Podsakoff, P. M., MacKenzie, S. B., Lee, J. Y., & Podsakoff, N. P. (2003). Common method biases in behavioral research: a critical review of the literature and recommended remedies. Journal of applied psychology, 88(5), 879.

Laschinger, H. K. S., & Fida, R. (2014). A time-lagged analysis of the effect of authentic leadership on workplace bullying, burnout, and occupational turnover intentions. European Journal of Work and Organizational Psychology, 23(5), 739-753.

SPENCE LASCHINGER, H. K., & Finegan, J. (2008). Situational and dispositional predictors of nurse manager burnout: a time‐lagged analysis. Journal of Nursing Management, 16(5), 601-607.

Kilroy, S., Flood, P. C., Bosak, J., & Chênevert, D. (2017). Perceptions of high‐involvement work practices, person‐organization fit, and burnout: A time‐lagged study of health care employees. Human Resource Management, 56(5), 821-835.

Your comment:

Also, there seems to be a typo in the number of data collected for wave one.

Authors should start a sentence with words and not figures as did with 49.2% and 56.2%.

Our response:

Thank you very much for your advice. We have corrected the typo in the number of data collected for wave one. In addition, we have started the sentence with words and not figures. 

Please see Page 9 Line 245- Line 247:

In the first-wave survey (Time 1), participants were asked to report their academic workload, separation from school, fears of contagion, and necessary demographic information. In Time 1, 1072 completed questionnaires were returned.

Please see Page 10 Line 254- Line 256:

Nearly half of the participants said they wanted or very much wanted the school to start soon (49.2%). Of the 867 college students who had participated, 56.2% of them were concerned or very concerned about the possibility of contracting COVID-19 after the semester began.

Your comment:

Measures

Authors may want to revise this section incorporating the number of items per scale, the original developer (author) of each scale, original psychometric property, and how it was scored in this study. Much more importantly, the psychometric properties of these measures for this study.

Our response:

Thank you very much. We carefully revised this section. In the revised manuscript, we described the number of items per scale, the original developer of each scale, the reliability of the original scale, and the reliability of scale in our research. All items we used in the research has been provided in Appendix. In addition, we described that participants responded to items on a 7-point scale or 5-point scale. Specifically, the scale of physical and psychological health has its Chinese version, and thus we can directly use it in our research. The scales of academic workload, separation from school, and perceived stress have their English version, and thus we followed the translation/back-translation procedure to translate English to Chinese. Finally, we developed the scale of fears of contagion, and test its reliability and validity. Based on the results of Cronbach's α, we found that the psychometric properties of these measures for this study are acceptable. 

Please see Page 10 Line 259- Page 12 Line 294:

Measures

We translated the measures from English to Chinese following the commonly used translation/back-translation procedure. All measures are reported in Appendix A. Unless otherwise noted, participants responded to all items on a 7-point scale (from 1=strongly disagree to 7=strongly agree).

Academic workload

It was measured using the three-item scale developed by Hystad et al. [35]. Items included: “I am spending a lot of time thinking about how this semester’s grades could negatively affect my educational and career goals,” “I am worrying a great deal about the effect this semester’s grades will have on my future,” and “I find myself very concerned about the grades I am likely to receive this semester.” Cronbach’s alpha in their study was .85, which suggests that this scale has good reliability. In the current study, Cronbach's α of this variable is .883. 

Separation from school

Separation from school was measured using the attachment avoidance scale developed by Smith et al. [54]. The scale has fifteen-item. An example of a reworded scale item is “I find it difficult to allow myself to depend on my group.” Factor analytic results in their research suggested that the scale has good reliability. In our current study, we reworded to refer to participants' classmates rather than their social group. An example of a reworded scale item is “I find it difficult to allow myself to depend on my classmates.” Cronbach's α of this variable is

.929.

Fears of contagion

We developed a six-item scale to assess participants’ fears of COVID-19 infection. Six items included “In public, I don't care about touching the door handle without protection,” “In public, I don’t mind sitting in a chair that has just been sat on,” “In an elevator, I don’t mind pushing a button without protection,” “When I’m in a crowded place, I don’t worry about coronavirus from other people,” “I don't worry about infection if other people don't wear masks,” “Wearing a mask would make me feel safe.” In the current study, Cronbach's α of this variable is .842. The results of confirmatory factor analysis are shown that χ2 = 74.424; df = 8; RMSEA = .098; CFI =.982; TLI = .966; SRMR = .043. Thus, this scale has acceptable reliability and validity. 

Perceived stress

The 10-item Perceived Stress Scale [55] was used to measure the student’s stress level in the past month. Participants responded to the items on a 5-point scale (from1=never to 5=very often). A sample item is “In the last month, how often have you felt nervous and “stressed”?” In our research, Cronbach's α of this variable is .792. 

Physical and psychological health

The CHQ-12 of the Chinese version was used to measure physical and psychological health. The CHQ-12 has been widely used in Chinese populations. The 12 items included headaches, heart palpitations, chest pain or tightening, trembling or pins and needles, sleeplessness, nervousness, and hopelessness. Participants responded to the items on a 4-point scale (1= not at all, 4=more than usual). A higher score represented a more severe psychosocial impairment. In our research, Cronbach's α of this variable is .895.

Please see Page 24 Line 622- Page 26 Line 677:

Appendix

Academic workload

1. I am spending a lot of time thinking about how this semester’s grades could negatively affect my educational and career goals.

2. I am worrying a great deal about the effect this semester’s grades will have on my future.

3. I find myself very concerned about the grades I am likely to receive this semester.

Separation from school

1. I find it difficult to allow myself to depend on my group.

2. I sometimes worry that I will be hurt if I allow myself to become too close to my group.

3. I am nervous when my group gets too close.

4. My desire to feel completely at one sometimes scares my group away.

5. I prefer not to depend on my group or to have my group depend on me.

6. I often worry that my group does not really accept me.

7. I am comfortable not being close to my group.

8. I often worry my group will not always want me as a member.

9. I am somewhat uncomfortable being close to my group.

10. My group is never there when I need it.

11. I find it difficult to completely trust my group.

12. I find my group is reluctant to get as close as I would like.

13. I am not sure that I can always depend on my group to be there when I need it.

14. I sometimes worry that my group doesn't value me as much as I value my group.

15. I want to be emotionally close with my group, but I find it difficult to trust my group completely or to depend on my group.

Fears of contagion

1. In public, I don't care about touching the door handle without protection.

2. In public, I don’t mind sitting in a chair that has just been sat on.

3. In an elevator, I don’t mind pushing a button without protection.

4. When I’m in a crowded place, I don't worry about coronavirus from other people.

5. I don't worry about infection if other people don't wear masks.

6. Wearing a mask would make me feel safe.

Perceived stress

1. In the last month, how often have you been upset because of something that happened unexpectedly? 

2. In the last month, how often have you felt that you were unable to control the important things in your life? 

3. In the last month, how often have you felt nervous and “stressed”? 

4. In the last month, how often have you felt confident about your ability to handle your personal problems? 

5. In the last month, how often have you felt that things were going your way? 

6. In the last month, how often have you found that you could not cope with all the things that you had to do? 

7. In the last month, how often have you been able to control irritations in your life? 

8. In the last month, how often have you felt that you were on top of things? 

9. In the last month, how often have you been angered because of things that were outside of your control? 

10. In the last month, how often have you felt difficulties were piling up so high that you could not overcome them?

Physical and psychological health

Have you recently... 

1. been suffering from headache or pressure in your head? 

2. had palpitation and worried that you might have heart trouble? 

3. had discomfort or a feeling of pressure in your chest? 

4. been suffering from shaking or numbness of your limbs? 

5. lost much sleep through worry? 

6. been taking things hard? 

7. been getting along well with your family or friends? 

8. been losing confidence in yourself? 

9. been feeling nervous and strung-up all the time? 

10. been feeling hopeful about your future? 

11. been worried about your family or close friends? 

12. felt that life is entirely hopeless?

Your comment:

Data analysis

Authors may end the method section with the above sub-title. This section may detail the types of analytic tools used and the purposes for using them. Cut-off or other criteria related to these analytic tools may be written here.

Our response:

Thank you very much for your advice. Following your suggestion, we have added the sub-title and provide more details about the types of analytic tools used, the purposes for using them and so on. 

Please see Page 12 Line 297- Line 314:

Analysis strategy

First, we conducted a series of confirmatory factor analyses (CFA) in order to test the discriminant validity of the five prime constructs (academic workload, separation from school, fears of contagion, perceived stress, and health). CFA, as an empirical research technology, is affiliated with structural equation modeling. Therefore, it is necessary to judge the fitting situation according to the fitting indexes from the structural equation model. Common fitting indexes include chi-square(χ2), degree of freedom (df), CFI, TLI, RMSEA, and SRMR. Specifically, if the ratio of chi square to degree of freedom is less than 5, the model is generally acceptable. When CFI and TLI are higher than 0.9, the model fits well. The smaller the RMSEA and SRMR, the better the result, and in particular, when it is below 0.08, the model is acceptable. Second, we performed Harman’s single-factor test to explore the potential influence of common method variance. No single factor accounting for more than 50% of the variance of all the relevant items indicates that the results are acceptable. Third, we calculated Pearson’s correlation coefficient using SPSS Version 21, which reflects the effect of change in one variable when the other variable changes. Fourth, we tested our hypotheses using a path analysis in Mplus Version 8.3. To examine mediation (Hypothesis 5,6, and 7), we used a bootstrap simulation with 1,000 replications to create our bias-corrected 95% confidence intervals (CIs) around our indirect effects. The bootstrap approach is a more robust strategy than the causal step procedure for small samples for assessing indirect effects and a useful method for avoiding power problems relating to a non-normal sampling of the indirect effect. When the 95% confidence interval of the path coefficient does not contain zero, the mediating effect is significant.

Your comment:

Results

Tests for common method variance: The first sentence under this sub-section seems incomplete.

Table 3: for bivariate relationships, authors may take out categorical variables from the results as it does not make meaning.

Table 4: Authors may add further details to the table as effect and CI are not enough to get a better picture of the analysis/data.

Our response:

Thank you very much for your advice. First, we revised the first sentence under this sub-section and make it complete. Second, we deleted the section of categorical variables in Table 3. Third, we provide more details about the analysis/data. 

Please see Page 13 Line 332- Line 338:

Tests for Common Method Variance

Because we collected student’s self-report of fears of contagion, separation from school, academic workload, and perceived stress and health in the study, common method variance (CMV) may present as a problem. Therefore, we measured different constructs at different time points to decrease common method variance as much as possible [56] and showed high respect for the security, anonymity, and privacy of research objects and informants. In addition, the results of Harman’s single-factor test suggest that an exploratory factor analysis of all items explained 68.86% of the total variance, and the largest factor accounted for only 24.73% of the variance.

Please see Page 14 Line 348- Line 349:

Please see Page 15 Line 372- Line 374:

Your comment:

Discussion

Page 14: “Given the ongoing COVID -19 epidemic to inducing panic, and anxiety [49], therefore, timely and…” this sentence is difficult to understand, please revise.

Our response:

Thank you very much for your advice. We have revised this confused sentence.

Please see Page 15 Line 375- Line 388:

Discussion

The spread of COVID-19 is becoming unstoppable and has already influenced people and countries all over the world. Holmes et al. [57] called for that multidisciplinary science research must be central to the international response to the COVID-19 pandemic and provide evidence-based guidance on responding to promoting people’s health and wellbeing during the COVID-19 pandemic. To answer this call, we focus on college students getting home-schooling to explore their stress and health problems. Although the COVID-19 is still spreading rapidly and widely worldwide, it has been effectively controlled in China. What has happened in China shows that quarantine, social distancing, and isolation of infected populations can contain the epidemic. Whereas individual coping strategies are possible (e.g., social distancing), the spread of the virus at a state level is still beyond any given individual’s control. The continuous spread of the epidemic, strict isolation measures, and delays in starting schools, colleges, and universities across the country are expected to influence college students. Considering stress and anxiety associated with the current COVID-19 pandemic for college students [49], it is urgent for the society and management departments to understand the actual situation of college students timely and accurately.

Your comment:

Authors may try to relate the theory to the findings thoroughly.

Our response:

Thank you very much for your advice. Following your suggestion, we integrated theory with our findings thoroughly in the revised manuscript. 

Please see Page 15 Line 375- Page 17 Line 432:

Discussion

The spread of COVID-19 is becoming unstoppable and has already influenced people and countries all over the world. Holmes et al. [57] called for that multidisciplinary science research must be central to the international response to the COVID-19 pandemic and provide evidence-based guidance on responding to promoting people’s health and wellbeing during the COVID-19 pandemic. To answer this call, we focus on college students getting home-schooling to explore their stress and health problems. Although the COVID-19 is still spreading rapidly and widely worldwide, it has been effectively controlled in China. What has happened in China shows that quarantine, social distancing, and isolation of infected populations can contain the epidemic. Whereas individual coping strategies are possible (e.g., social distancing), the spread of the virus at a state level is still beyond any given individual’s control. The continuous spread of the epidemic, strict isolation measures, and delays in starting schools, colleges, and universities across the country are expected to influence college students. Considering stress and anxiety associated with the current COVID-19 pandemic for college students [49], it is urgent for the society and management departments to understand the actual situation of college students timely and accurately. Based on the Transactional Model of Stress and coping [49], this study explored the influence of academic workload, separation from school, and fears of contagion on college students’ physical and physiological health, as well as the mediating effect of perceived stress in those relationships. 

The current study contributes to the existing literature. First, the present study goes beyond previous literature on college students’ health during the epidemic by integrating three types of stressors from different fields in the proposed model. As highlighted in previous research on college students’ academic stress, preparing exams, courses, and papers should exhibit a negative effect on individual health. During the COVID-19 outbreak, Chinese college students’ learning was not suspended, but they attend the various courses offered online follow the regular schedule. While those measures of the virtual semester ensure regular study, they also cause stress on students. In addition, given the importance of social groups to an individual’s identity and self-worth, we found that college students were separated from their classmates during the COVID-19 epidemic, which brought them stress and anxiety. Previous evidence suggests that college students usually keep attachment relationships with their social group [12,58]. Attachment figures are usually parents, but may also be siblings, grandparents, or group. Unlike most previous studies that focused on separation from parents [9], this study focused on the influence of departure from school and schoolmates, which is particularly relevant to the epidemic situation. For college students attached to their school or classmates, school-closure is a kind of separation experience, which may be different from their experience when they leave home. Considering the relatively new separation (separation from school) caused by the outbreak of COVID-19, our findings suggest that separation from school was positively related to college students’ perceived stress during home-schooling. Finally, we found that exposure to a potentially infectious environment would lead to people’s stress, which is in line with previous research that pointed out the negative correlation between the risk of infection and life satisfaction [58]. Similar results have been found in the study of College Students’ psychological adjustment during SARS. For example, Main et al. [12] found that the experience of SARS-related stressors was positively associated with psychological symptoms for Chinese college students during the outbreak. Thus, we supposed that during an acute large-scale epidemic such as the SARS and COVID-19 epidemic, even among persons who were not directly contaminated with the disease, the psychological influence of the outbreak on them was significant. In doing so, we identified three important stressors for college students in the COVID-19 pandemic, providing essential inspiration for college students to maintain their physical and mental health during the current epidemic. 

Second, based on a transactional model view, we provide a plausible mechanism for explaining the association between academic workload, separation from school, and fears of contagion and health. The transactional model posits that stress responses emerge from appraisal processes that begin when individuals experience a stressor. During primary appraisal, perceptions of elements of the focal stressor are used to determine the degree of threat or harm that this stressor represents; during secondary appraisal, individuals consider if and how they can resolve the underlying stressor. COVID-19, a contagious respiratory illness, is an ongoing, global health crisis, and the greatest challenge we have faced since World War II [59]. The COVID-19 pandemic is a grim but illustrative anxiety-inducing stressor; an uncertain and ongoing threat that cannot be resolved via individual efforts. When individuals have few resources, ways, or abilities at their disposal to deal with the stressors, they generate stress and anxiety and ultimately lead to negative consequences. Thus, perceived stress may be a mediator, transmitting the effects of academic workload, separation from school, and fears of contagion on health-related outcomes. These findings suggest that academic workload, separation from school, and fears of contagion may contribute to youth’s general perceived stress, which in turn, may negatively influence their physical and psychological health. Our findings supported Lorenzo-Blanco and Unger’s [60] and Sariçam’s [61] proposition that perceived stress plays an important role in influencing psychological well-being. 

Your comment:

Explaining the “SFS” with separation anxiety disorder is too far-fetched (exaggerated and unconvincing). Besides, these are students above 17 year old. SFS have a role to play in the lives of students but not from the angle of separation anxiety but probably as a supporting/coping strategy.

Our response:

Thank you very much for your advice. Following your suggestion, we explained “SFS” from the point of attachment in the revised manuscript. From a supporting/coping strategy, we provided some advice to help college students keep healthy when they separate from school. 

Please see Page 16 Line 397- Line 406:

In addition, given the importance of social groups to an individual’s identity and self-worth, we found that college students were separated from their classmates during the COVID-19 epidemic, which brought them stress and anxiety. Previous evidence suggests that college students usually keep attachment relationships with their social group [12,58]. Attachment figures are usually parents, but may also be siblings, grandparents, or group. Unlike most previous studies that focused on separation from parents [9], this study focused on the influence of departure from school and schoolmates, which is particularly relevant to the epidemic situation. For college students attached to their school or classmates, school-closure is a kind of separation experience, which may be different from their experience when they leave home. Considering the relatively new separation (separation from school) caused by the outbreak of COVID-19, our findings suggest that separation from school was positively related to college students’ perceived stress during home-schooling.

Your comment:

Limitation

The way the authors reported the procedure of the data collection did not reflect a cross-sectional design. Hence, it will make readers doubt the authenticity/validity of this study.

Our response:

Thank you very much for your advice. We revised this point in the revised manuscript. Although we used a time-lagged design to collect data at different times, future research can conduct the experimental design or utilize the longitudinal data to ensure the conclusion reflects causation.

Please see Page 18 Line 445- Page 18 Line 458:

Limitations

Despite the potential contribution that the present study makes to the mental health field, limitations of the study should be noted. First, a potential limitation is that all measures came from the same source, raising the potential for same-source measurement biases. However, we used a variety of means to reduce this issue, including varying our response scales and separating our measures in time. Further, as we were interested in how college students have dealt with the pandemic over time, focusing on self-reported experiences was appropriate. Another potential limitation of our study is concerned with causality. Hence, future research should conduct the experimental design or utilize the longitudinal data to ensure the conclusion reflects causation. Finally, our research reflects only the impact of the COVID-19 pandemic for college students, and much work is needed to gain a complete understanding of the implications of this crisis for students. Future research could consider how individual factors, such as self-concept, and contextual factors, such as social support, may influence college students’ response to perceived stress. Moreover, research focused on within-person fluctuations of perceived stress during this time would also be instructive, as there is no doubt that individuals have experienced considerable variability on a daily basis during the pandemic.

---

## [Decision Letter · Decision Letter 1]

25 Jan 2021

College students’ stress and health in the COVID-19 pandemic: the role of academic workload, separation from school, and fears of contagion

PONE-D-20-32875R1

Dear Dr. Yang,

We’re pleased to inform you that your manuscript has been judged scientifically suitable for publication and will be formally accepted for publication once it meets all outstanding technical requirements.

Kind regards,

Chung-Ying Lin

Academic Editor

PLOS ONE

Additional Editor Comments (optional):

Dear authors, 

Thank you for addressing the reviewer's comments. There is only on minor issue remains and I believe that you can deal with it during the proof stage.

Reviewers' comments:

Reviewer's Responses to Questions

**Comments to the Author**

1. If the authors have adequately addressed your comments raised in a previous round of review and you feel that this manuscript is now acceptable for publication, you may indicate that here to bypass the “Comments to the Author” section, enter your conflict of interest statement in the “Confidential to Editor” section, and submit your "Accept" recommendation.

Reviewer #1: All comments have been addressed

2. Is the manuscript technically sound, and do the data support the conclusions?

Reviewer #1: Yes

3. Has the statistical analysis been performed appropriately and rigorously? 

Reviewer #1: Yes

4. Have the authors made all data underlying the findings in their manuscript fully available?

Reviewer #1: Yes

5. Is the manuscript presented in an intelligible fashion and written in standard English?

Reviewer #1: Yes

6. Review Comments to the Author

Reviewer #1: The authors have comprehensively revised the manuscript to my satisfaction. A minor comment though.

Authors should make sure that all p-values are exact (e.g., p=.032) than stating it as "p<.05".

7. PLOS authors have the option to publish the peer review history of their article (what does this mean?). If published, this will include your full peer review and any attached files.

Reviewer #1: No

---

## [Editor Report · Acceptance letter]

1 Feb 2021

PONE-D-20-32875R1 

College students’ stress and health in the COVID-19 pandemic: the role of academic workload, separation from school, and fears of contagion 

Dear Dr. Yang:

I'm pleased to inform you that your manuscript has been deemed suitable for publication in PLOS ONE. Congratulations! Your manuscript is now with our production department. 

Kind regards, 

on behalf of

Dr. Chung-Ying Lin 

Academic Editor

PLOS ONE